# Evaluation of Safe Insertion Angles for Spinal Needles and Safe Intensity of the Holmium:YAG Laser during Percutaneous Laser Disc Ablations in Feline Cadavers

**DOI:** 10.3390/vetsci11070325

**Published:** 2024-07-18

**Authors:** Zhenglin Piao, Young-ung Kim, Jongchan Ko, Jumjae Lee, Daeyoung Choi, Namsoo Kim

**Affiliations:** 1Department of Veterinary Medicine, College of Agriculture, Yanbian University, Yanji 133002, China; 0000008765@ybu.edu.cn; 2Yeon Animal Medical Center, Seoul 07700, Republic of Korea; blue03sky87@gmail.com; 3Department of Veterinary Surgery, College of Veterinary Medicine, Jeonbuk National University, Iksan 54596, Republic of Korea; chan1207s@jbnu.ac.kr (J.K.); petdoctor7500@naver.com (J.L.); panda073@naver.com (D.C.)

**Keywords:** intervertebral disc disease, minimally invasive techniques, Holmium:YAG (Ho:YAG) laser

## Abstract

**Simple Summary:**

Spinal surgeries in cats often require precise and minimally invasive methods to ensure successful outcomes with minimal complications. This study focused on using a specialized laser, known as the Holmium:YAG (Ho:YAG) laser, to perform a type of spinal surgery that involves less cutting and is generally safer for animals. The main goal was to find the best settings for the laser and the safest way to insert it during surgery, thus ensuring it effectively treats the problem without harming surrounding areas. Through using a technique called computed tomography, or CT scans, the researchers were able guide the surgery very accurately. The results show that a specific setting on the laser was optimal for treating spinal issues in cats while keeping the surrounding tissues safe. These findings are important as they can lead to safer, more effective spinal treatments for cats, thus benefiting veterinary care by providing a reliable method that could be widely used to help animals with similar problems.

**Abstract:**

In the context of veterinary medicine, minimally invasive techniques for feline spinal surgery remain underexplored, particularly for percutaneous laser disc ablation (PLDA) when using the Holmium:YAG (Ho:YAG) laser. This study aimed to refine the application of the Ho:YAG laser in PLDA by determining the optimal laser intensity and safe insertion angles for the thoracic and lumbar intervertebral discs (IVDs) in cats. Through utilizing computed tomography (CT) for precise guidance, this research involved a cadaveric study of 10 cats to evaluate the spatial configurations that allow for safe needle insertions and effective laser ablation. Various energy settings of the Ho:YAG laser (20 J, 40 J, and 60 J) were tested to ascertain the balance between adequate disc vaporization and minimal adjacent tissue damage. The results demonstrate that a 40 J setting is the most effective in achieving significant disc decompression without compromising surrounding tissue integrity. Additionally, the CT scans proved crucial in confirming the accuracy of the needle placement and the safety of the laser application angles. This study established that the 40 J setting on the Ho:YAG laser, combined with CT-guided insertion techniques, offers a reliable method for PLDA, thus enhancing the safety and efficacy of feline spinal surgeries.

## 1. Introduction

IVDD is a pathological condition in which the central abnormal nucleus portion herniate or protrude through the weakened portion of the outer rings [1]. A protruded portion of the disk can compress the adjacent spinal nerves and result in severe injury, neurologic pain, and abnormalities in propagating action potentials. Some studies have described the prevalence, imaging features, correction, and clinical outcomes of clinically significant feline IVDD [2,3,4,5,6,7,8,9,10,11,12,13,14]; however, there is a lack of available data on feline IVDD when compared to dog and human IVDD. In a previous study, thoracic lumbar IVDD showed a shorter duration of clinical signs (mean: 4 days vs. 72 days), with more severe neurological deficits [5]. It has been suggested that the incidence of IVDD in cats may be more frequent than that which has been currently reported, though without having any clinical relevance [8]. Furthermore, spinal fracture and luxation are also common neurologic and orthopedic clinical problems in cats [15]. Spinal stabilization is recommended for the treatment of vertebral column disorders, such as traumatic spinal fractures or luxations, and vertebral instability after spinal decompression surgery.

Surgical techniques to correct the IVDD in cats include hemilaminectomy [2], mini-hemilaminectomy, dorsal laminectomy [3], and corpectomy [11] or disk fenestration [7]. However, over the past two decades, minimally invasive surgical approaches have become efficient and popular for the management of various spinal disorders, thus demonstrating numerous advantages over the traditional open approaches in human [16,17] and veterinary medicine [18,19].

Despite satisfactory outcomes having been reported with modern surgical techniques, these techniques are technically challenging and are associated with relatively high mortality and failure rates (about 20%) [20]. The small size of corridors in affected dogs compared to human and narrower bone corridors, which are used to position stabilizing implants, are often responsible for the major technical problems of these procedures in dogs [21], and these techniques are the most difficult in cats.

In human medicine, extensive precautions and preoperative preparations, such as cadaveric studies, advanced imaging data, and various imaging-based intraoperative guidance, are considered to avoid vertebral canal violation and vertebral artery injury during stabilization [22,23,24]. In veterinary spinal surgery, particularly in feline spinal surgery, there is a lack of such types of preoperative and intraoperative data, which are important for achieving successful outcomes. In addition, the selection of the type of laser, wavelength, and energy is critical in laser ablation for disc diseases. Our initial objective was to provide data on the safe corridor of spinal needle insertion via a cadaveric study of thoracolumbar and lumbosacral IVDs in order to determine the appropriate intensity and total energy for laser disk ablation when using the Ho:YAG laser.

## 2. Materials and Methods

### 2.1. Preparation of the Vertebral Bodies in Cadavers

In this cadaveric study, 7 cats (mean weight: 4.12 kg; range, 3.5–5.3 kg) with no abnormalities in their IVDs or vertebrae, as well as an absence of spinal cord diseases, were euthanatized. The characteristics of the cadavers, including their breed, sex, and weight, were recorded (Table 1) before conducting radiography examination, following which the cadavers were preserved at −21 °C.

### 2.2. Spinal Needle Safe Corridor Evaluation

In the transverse view of the CT images, the IVDs from the 10th thoracic vertebra (T10) to the seventh lumbar vertebra (L7) and the 10th vertebra to the first sacral vertebra were evaluated. The CT images were evaluated using a picture archiving and communication system (INFINITT PACS, INFINITT Healthcare Co., Seoul, Republic of Korea). The CT scanner used was the Alexion TSX-034A scanner (Canon Medical Systems; Tokyo, Japan). The transverse view of the CT image at each IVD location was evaluated to determine the best position for inserting the spinal needle safely, thus avoiding damage to the surrounding structures (the bone and parenchyma). The center of each IVD was the point at which the longest transverse axis meets the longest longitudinal axis. Using CT images, the exact position of the intervertebral disc was determined. The intersection of the longest transverse and longitudinal axes of the disc was identified, and the angle from the horizontal plane of the transverse view to the accessory process or caudal articular process was measured to determine the safe insertion angle (Figure 1 and Figure 2).

### 2.3. Safe Corridor Evaluation at the Thoracic Region

The insertion direction of the spinal needle was considered dorsolateral from T10 to L1. At the center of the thoracic IVD, the angle required to avoid the accessory process and the parenchymal organ was measured based on the dorsal plane. The insertion point for the safe corridor angle measurement was the rib for the thoracic vertebrae and the dorsal plane for the lumbar vertebrae, with the ending point at the accessory process or caudal articular process, which ranges from the T10 to L1 intervertebral discs. In the thoracic vertebrae, measurements were mainly taken from the ribs to the accessary process (T10–L1) (Table 2).

### 2.4. Safe Corridor Evaluation at the Lumbar Region

At the lumbar IVD, a spinal needle was inserted via the lumbar junction. The insertion and ending points for the safe corridor angle measurements ranged from the L1 to the L7 intervertebral discs. In the lumbar vertebrae, measurements were mainly taken from the dorsal plan to the accessary process (L1–7) (Table 2).

### 2.5. Macroscopic Evaluation According to Laser Intensity

We collected the T10–L7 intervertebral discs of the seven cat cadavers (Figure 1). The soft tissues around the thoracolumbar vertebra were removed to expose the IVDs. The center of each vertebral body was transversely sectioned using an oscillating saw, and the IVD was placed centrally for laser application. Each segment was weighed before the procedure using an electronic scale (XT220A, Precisa, Switzerland). Each specimen was tested on a waterproof cloth while being weighed to eliminate external influences before and after measurement. Subsequently, the spinal needle was inserted in the dorsolateral direction in the thoracolumbar IVD (Figure 3 and Figure 4). To determine whether the tip of the spinal needle was located in the NP, the spinal needle was inserted toward the center of the IVD until there was a reduction in the spinal needle pressure and the stylet of the spinal needle was pushed back by the pressure exerted by the NP. We used the Ho:YAG laser (HOLIN-WON30^®^, WON TECH Co., Ltd., Seongnam, Republic of Korea). The laser fiber was checked in advance by attaching a 2 mm equivalent to the tip of the spinal needle before insertion. The total energy was divided into 20 J, 40 J, and 60 J (Table 1). The other conditions, including the 2 W (0.4 J/pulse, 5 Hz) energy output and 21 G spinal needle, were the same for all IVDs.

### 2.6. Cadaver Preparation and Computed Tomography

Ten cadavers (mean weight: 3.97 kg, range: 3.5–4.3 kg) of cats, who were euthanatized for reasons unrelated to this study, had no abnormalities in the IVD and vertebrae, and with an absence of spinal cord disease, were included in this study. The characteristic features of the cadavers, including breed, sex, and weight, were recorded, and CT was performed. CT images were obtained using a 16-row multi-detector CT scanner (Alexion, TSX-034A, Toshiba Medical System, Tochigi, Japan) with the following parameters: 120 kVp, 150 mAs, 0.688 pitch, 0.75 rotation time, and 0.5 mm slice thickness. All the cross-sectional images had undergone multi-planar reconstruction with an interval of 0.1 mm using a bone algorithm. After the CT, the cadavers were stored at −21 °C.

### 2.7. Vaporized NP Weight According to the Total Energy

Ten cat cadavers were prepared as described above. The total energy was divided into 20 J, 40 J, and 60 J. The other conditions, including a 2 W (0.4 J/pulse, 5 Hz) energy output and 21 G spinal needle, were the same for all IVDs.

### 2.8. Histopathological Analysis

The IVDs, including endplates, were harvested and sagittally cut into two parts from the middle of the disc for histological study. Specimens were immersed in 4% formaldehyde for 1 week and decalcified using 20% EDTA for 1 month. Then, 6 μm-thickness dehydrated and paraffin-embedded slides were made. The slides were stained with hematoxylin and eosin for cellular constituents, as was described previously [25].

### 2.9. Statistical Analysis

D’Agostino and Pearson omnibus normality tests were used to determine the normal distribution of the data. A comparison t-test was used to determine the level of correlation between the outcomes before and after consideration of the segment weight, and *p* < 0.05 was considered statistically significant. As the decreased NP weight of all groups was normally distributed, one-way analysis of variance was used, and *p* < 0.05 was considered statistically significant. A Bonferroni post hoc test was conducted to determine the statistical difference between each measured value. All analyses were performed using Prism 5.03 software (GraphPad Software Inc., San Diego, CA, USA).

## 3. Results

### 3.1. Safe Corridor Angle at the Thoracolumbar Region

We examined 10 cadavers and measured the safe corridor angle from the T10–T11 to the L6–L7 IVDs. The mean ± standard deviation (SD) starting safe corridor angle at T10–T11, T11–T12, T12–T13, and T1–L1 was 7.09° (±1.55), 9.72° (±2.57), 10.60° (±3.07), and 8.51° (±3.03), respectively, and the finishing angle was 34.43° (±2.83), 37.25° (±2.85), 37.14° (±4.29), and 39.12° (±2.09), respectively (Table 3). In the L1–L7 IVDs, all the safe corridor angles at L1–L2, L2–L3, L3–L4, L4–L5, L5–L6, and L6–L7 started at the dorsal plane, and the finishing angle was 41.91° (±1.52), 42.41° (±2.25), 42.51° (±1.20), 42.24° (±1.47), and 43.00° (±1.09), respectively (Table 4).

### 3.2. NP Changes after Laser Treatment

The gross morphological changes after three types of laser treatment were evaluated and compared using gross photographs and CT images. Through using gross photographs, it was visually observed that the tissues were least damaged in the 20 J-treated group, followed by the 40 J-treated groups. The tissues in the 60 J-treated group were severely damaged compared to those in the 20 J- and 40 J-treated groups (Figure 5). For more confirmation, CT (Figure 6) images were also compared; however, it was still not clear. Therefore, histopathological analysis was performed, which showed that the surrounding tissues in the 20 J- and 40 J-treated groups were slightly damaged; however, those in the 60 J-treated group were severely damaged (Figure 7).

### 3.3. Vaporized NP Weight According to Total Energy

A total of 70 IVDs from seven cat carcasses (10 IVDs from each cat) were selected for laser intensity and energy tests. There were three groups according to the intensity of laser energy used—the 20 J group (20 IVDs); the 40 J group (30 IVDs), and the 60 J group (20 IVDs). The other conditions, including a 2 W (0.4 J/pulse, 5 Hz) energy output and 21 G spinal needle, were the same for all IVDs. At a total energy of 20 J, the average weights of the NP before and after application of the laser were 3470.40 mg (±228.57) and 3452.15 mg (±229.09), respectively. The average weight of the NP before and after application of a total energy of 40 J was 4025.73 mg (±181.09) and 3998.97 mg (±180.90), respectively. The average weight of the NP before and after the application of a total energy of 60 J was 3950.75 mg (±212.18) and 3912.25 mg (±211.51), respectively, and the average of the reduced vaporized NP weight was 18.25 mg (±7.08). In all experiments, the vaporized NP weight demonstrated a significant degree of reduction before and after the experiment (*p* < 0.001). The difference in the NP weight before and after the laser application of a total energy of 20 J, 40 J, and 60 J was 18.25 mg (±9.63), 26.77 mg (±14.50), and 37.50 mg (±13.22), respectively. The difference in the NP weight in the 40 J-treated (*p* < 0.05) and 60 J-treated (*p* < 0.001) groups significantly differed from that in the 20 J-treated group (Table 5, Figure 8).

## 4. Discussion

In feline surgery, there is a lack of accurate data such as those on corridor angle, implant positioning, or the selection of appropriate techniques for feline spine surgical stabilization. A safe corridor angle is important for minimally invasive spine surgery. Further, the appropriate type of laser, laser wavelength, and energy is crucial in laser ablation for spinal disc diseases. In general, it has been recommended to follow the same techniques as those used for dogs [26]. However, the anatomical structures of cats are different from those of dogs; therefore, stabilization with pins/screws and polymethylmethacrylate corridors are a commonly used construct, but the information on the challenges involved in the procedure for cats is still lacking. To the best of our knowledge, there have been no cadaveric studies on the measurement of the thoracolumbar corridor angle in cats.

Our results show that the safe corridor width (W) is quite narrow in cats. The mean ± SD starting safe corridor angle at T10–T11, T11–T12, T12–T13, and T13–L1 was 7.09° (±1.55), 9.72° (±2.57), 10.60° (±3.07), and 8.51° (±3.03), respectively, and the finishing angle was 34.43° (±2.83), 37.25° (±2.85), 37.14° (±4.29), and 39.12° (±2.09), respectively. In the L1–L7 IVDs, all of the safe corridor angles at L1–L2, L2–L3, L3–L4, L4–L5, L5–L6, and L6–L7 started at the dorsal plane, and the finishing angle was 41.91° (±1.52), 42.41° (±2.25), 42.24° (±1.47), and 43.00° (±1.09), respectively. The starting and finishing angles differed at each point of the vertebral space. Feline vertebral bodies have an hourglass-like shape, and their transverse section varies in shape from rectangular to triangular to oval. The vertebral body length in the thoracic vertebrae is shorter than that in the lumbar vertebrae. The middle section of the vertebral body is smaller than the cranial and caudal sections in the thoracic and lumbar vertebrae [27]. Therefore, the variation in the safe corridor angle may be due to the anatomical variation in the feline vertebral collum.

A percutaneous technique for the vaporization or photothermal ablation of the NP in lumbar discs in humans using laser energy has been reported as a treatment of IVDD since 1975 [28], albeit with some difficulty as a transmission of the respective wavelengths through optical fibers is not possible or extremely restricted owing to technologic limitations. Unlike the information on safe corridors, the data on the appropriate type of laser, wavelength, and energy in cats are scarce. Many types of laser are used for laser ablation procedures such as carbon dioxide (CO_2_—λ = 10.6 μm), diode (λ = 0.805/0.980 μm), excimer laser (λ = 0.308 μm), neodymium–yttrium aluminum garnet (Nd:YAG—λ = 1.064 μm), and holmium– yttrium aluminum garnet (Ho:YAG—λ = 2.1 μm) [28]. All lasers have some advantages and disadvantages; however, the Ho:YAG laser has particular advantages over the other approved lasers [29,30,31] in terms of disc ablation. The wavelength of Ho:YAG is strongly absorbed by water, the depth of tissue penetration is minimized, and the area of necrosis and collateral thermal effects are limited owing to the high water content of the NP. The Ho:YAG laser is also a pulsed laser (5–12 Hz), which allows for a cooling of tissue between pulses, thus potentially limiting tissue damage through the thermorelaxation phenomenon [32]. Therefore, we chose the Ho:YAG laser in this experiment. Furthermore, data on the appropriate energy and the reduction in the NP weight in cats are insufficient. Therefore, we used three different energies (20, 40, and 60 J) of the Ho:YAG laser for disc ablation and measured the NP weight in cat cadavers and compared them. To the best of our knowledge, no studies have used the Ho:YAG laser, and none have performed a comparative study to evaluate the changes in the NP weight after laser ablation using cat cadavers.

In this study, a total of 70 IVDs from seven cat carcasses (10 IVDs from each cat) were selected for laser intensity and energy tests. The IVDs were divided into three groups according to the laser energy—the 20 J-treated group (20 IVDs); the 40 J-treated group (30 IVDs); and the 60 J-treated group (20 IVDs). At a total energy of 20 J, the average weight of the NP before and after the application of the laser was 3470.40 ± 228.57 and 3452.15 ± 229.09, respectively. The average weight of the NP before and after the application of a total energy of 40 J was 4025.73 mg (±181.09) and 3998.97 mg (±180.90), respectively. The average weight of the NP before and after the application of a total energy of 60 J was 3950.75 mg (±212.18) and 3912.25 mg (±211.51), respectively, and the average of the reduced vaporized nuclear medulla weight was 18.25 mg (±7.08). In all experiments, the vaporized NP weight demonstrated a significant degree of reduction before and after the experiment (*p* < 0.001). The difference in the NP weight before and after the laser application of a total energy of 20 J, 40 J, and 60 J were 18.25 mg (±9.63), 26.77 mg (±14.50), and 37.50 mg (±13.22), respectively. The difference in the NP in the 40 J-treated (*p* < 0.05) and 60 J-treated (*p* < 0.001) groups significantly differed from that in the 20 J-treated group.

The gross morphological changes after three types of laser treatment were evaluated and compared by gross photographs and CT images. Using gross photographs, it was visually observed that the tissues were the least damaged in the 20 J-treated group, followed by the 40 J-treated group. However, the tissues in the 60 J-treated group were severely damaged compared to those in the 20 J- and 40 J-treated groups. For more confirmation, the CT images were also compared; however, the changes induced by different types of laser energy by these images were not diagnosed clearly. Therefore, histopathological analysis was also performed, which showed that the surrounding tissues in the 20 J- and 40 J-treated groups were slightly damaged; however, those in the 60 J-treated group were severely damaged. Therefore, a 40 J laser might be safe and effective to correct the NP.

## 5. Conclusions

Extensive precautions and preoperative preparations such as cadaveric studies, advanced imaging data, and various imaging based intraoperative guidance were considered to avoid vertebral canal violation and vertebral artery injury during stabilization. In veterinary spinal surgery, particularly in feline spinal surgery, there is a lack of preoperative and intraoperative data, which are important for achieving successful outcomes. Our initial objective was to provide data on the safe corridor of spinal needle insertion via a cadaveric study of thoracolumbar and lumbosacral IVDs in cats. Based on the above results and discussion, this study provides a foundation for the clinical application of safe corridor spinal surgery. The mean ± SD starting safe corridor angle at T10–T11, T11–T12, T12–T13, and T13–L1 was 7.09° (±1.55), 9.72° (±2.57), 10.60° (±3.07), and 8.51° (±3.03), respectively, and the finishing angle was 34.43° (±2.83), 37.25° (±2.85), 37.14° (±4.29), and 39.12° (±2.09), respectively. In the L1–L7 IVDs, all the safe corridor angles at the L1–L2, L2–L3, L3–L4, L4–L5, L5–L6, and L6–L7 IVDs started at the dorsal plane, and the finishing angle was 41.91° (±1.52), 42.41° (±2.25), 42.24° (±1.47), and 43.00° (±1.09), respectively. In addition, selection of the type of laser, wavelength, and energy is critical in laser ablation for disc diseases. The Ho:YAG laser is effective and safe for decompressing herniated IVDs under the conditions of the feline cadaveric models established in this study. The difference in the NP weight before and after the laser application of a total energy of 0 J, 40 J, and 60 J was 18.25 mg (±9.63), 26.77 mg (±14.50), and 37.50 mg (±13.22), respectively. The difference in the NP weight in the 40 J- and 60 J-treated groups significantly differed from that in the 20 J-treated group. Therefore, by considering the tissue damage and loss in NP weight, 40 J was considered the most effective and safe total energy value.

## Figures and Tables

**Figure 1 vetsci-11-00325-f001:**
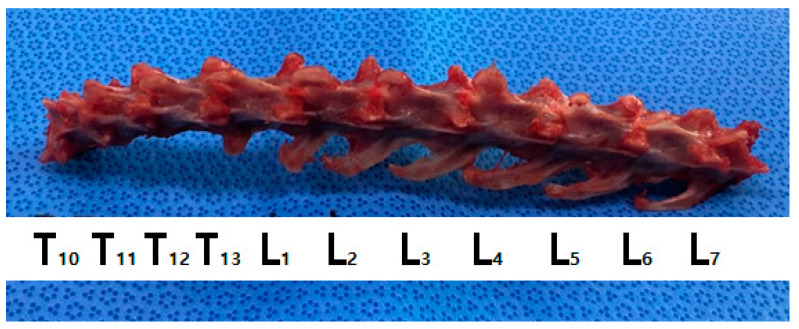
Collected vertebrae after dissection from a cat’s cadaver. From the T10 thoracic vertebra to the seventh lumbar vertebra (L7).

**Figure 2 vetsci-11-00325-f002:**
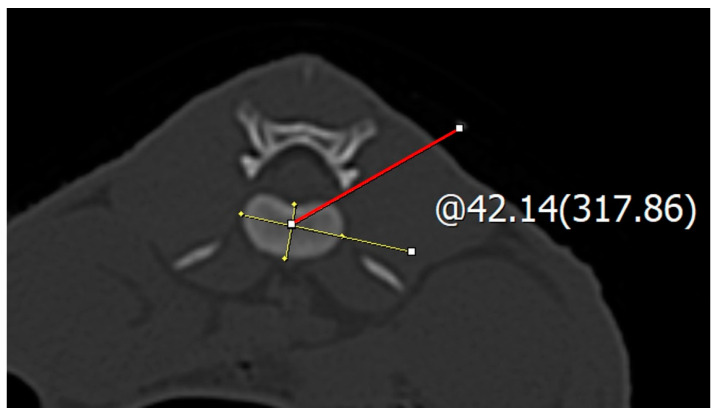
The measurement of the safe corridor angle in a L1–2 intervertebral disc on a transverse CT image. The red line represents the safe corridor.

**Figure 3 vetsci-11-00325-f003:**
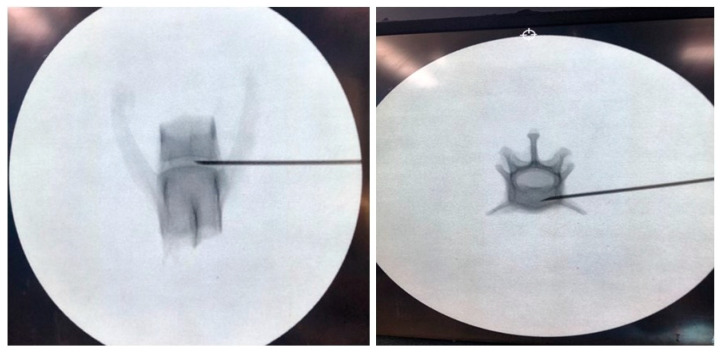
The safe insertion of the needle to the same correct position before laser treatment was confirmed with the C-RAM.

**Figure 4 vetsci-11-00325-f004:**
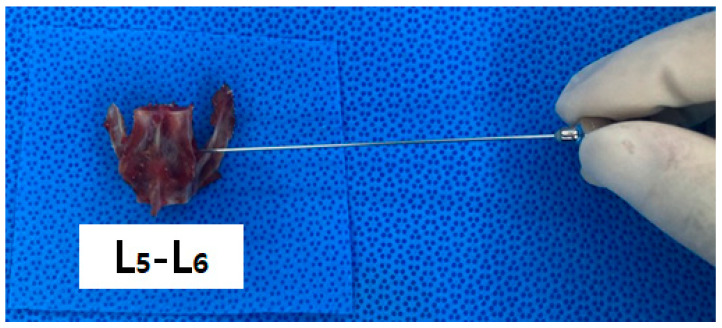
Insertion of the spinal needle and laser fiber into the intervertebral disc space via placing the segment on the waterproof cloth.

**Figure 5 vetsci-11-00325-f005:**
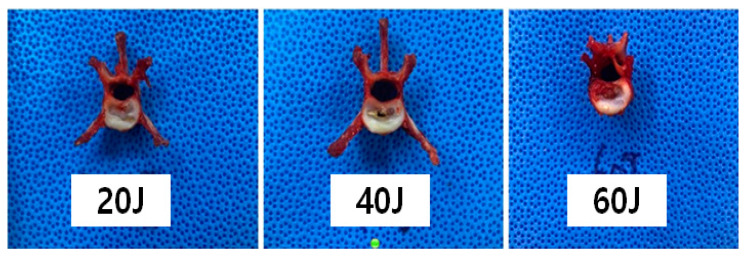
Macroscopic evaluation of the amount of intervertebral disc damage after three different types of laser (20 J, 40 J, and 60 J) treatment. The tissue of the 60 J-treated group were severely damaged when compared with the 20 J- and 40 J-treated groups.

**Figure 6 vetsci-11-00325-f006:**
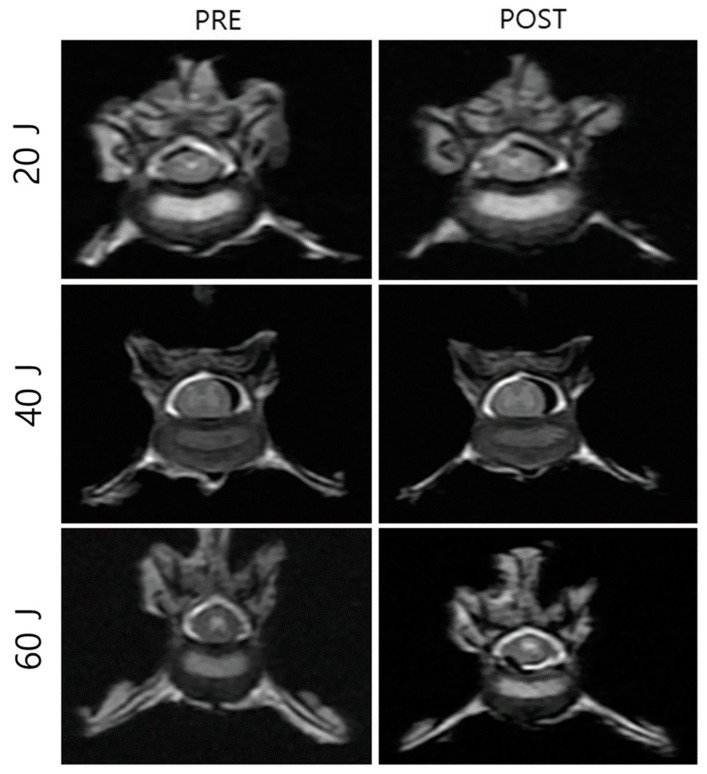
Pre- and post-laser treatment computed tomography images that were used to evaluate intervertebral disc damage after three different types of laser (20 J, 40 J, and 60 J) treatment. However, the NP and surrounding tissue damage was not clear.

**Figure 7 vetsci-11-00325-f007:**
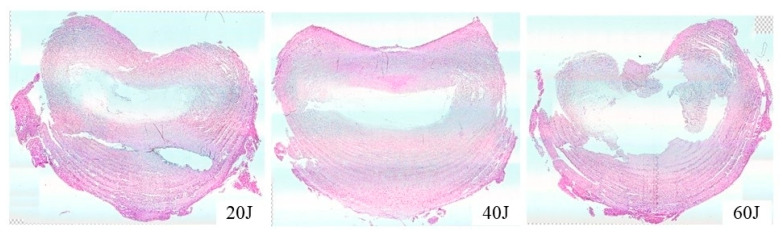
Microscopic evaluation of the amount of intervertebral disc damage after three different types of laser (20 J, 40 J, and 60 J) treatment. The surrounding tissue of the 20 J and 40 J groups was slightly damaged, but that of the 60 J-treated group was severely damaged. Therefore, a 40 J laser might be safe and effective for correcting the nucleus pulpous.

**Figure 8 vetsci-11-00325-f008:**
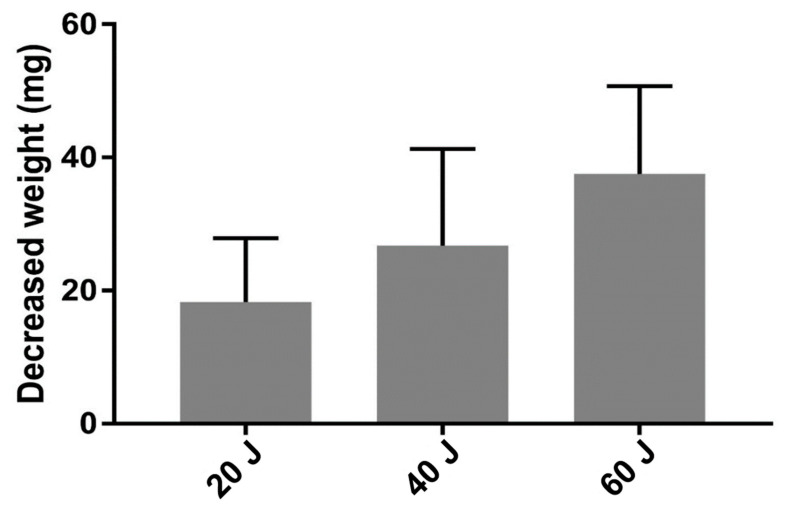
Analysis of the nucleus pulposus weight reduction according to the total energy. There was no significant decrease in the nucleus pulposus weight when under 60 J, 80 J, or 100 J conditions (*p* > 0.05).

**Table 1 vetsci-11-00325-t001:** Signalments of the cadavers.

No.	Breeds	Sex	Body Weight (kg)	Experiment	Total Energy
1	KSH *	Castrated male	3.5	Laser intensity and nucleus pulposus weight	20 J
2	KSH	Spayed female	4	Laser intensity and nucleus pulposus weight	20 J
3	KSH	Spayed female	4.2	Total energy and nucleus pulposus weight	40 J
4	KSH	Castrated male	3.8	Total energy and nucleus pulposus weight	40 J
5	KSH	Castrated male	4.1	Nucleus pulposus weight	40 J
6	KSH	Castrated male	4.3	Nucleus pulposus weight	60 J
7	KSH	Spayed female	3.9	Nucleus pulposus weight	60 J

* KSH: Korean short hair breed.

**Table 2 vetsci-11-00325-t002:** The insertion and ending points of the safe corridor angle measurement ranges of the T10–11 to L6–7 intervertebral discs.

No.	T10–11 IVDs	T11–12 IVDs	T12–13 IVDs	T13–L1 IVDs	L1–2 IVDs	L2–3 IVDs	L3–4 IVDs	L4–5 IVDs	L5–6 IVDs	L6–7 IVDs
1	Rib~Acc	Rib~Acc	Rib~Acc	Rib~Acc	DP~Acc	DP~Acc	DP~Acc	DP~Acc	DP~Cau	-
2	Rib~Acc	Rib~Acc	Rib~Acc	Rib~Acc	DP~Acc	DP~Acc	DP~Acc	DP~Acc	DP~Cau	-
3	Rib~Acc	Rib~Cau	Rib~Acc	Rib~Acc	DP~Acc	DP~Acc	DP~Acc	DP~Acc	DP~Acc	-
4	Rib~Cau	Rib~Acc	Rib~Acc	Rib~Acc	DP~Acc	DP~Acc	DP~Acc	DP~Acc	DP~Cau	-
5	Rib~Acc	Rib~Acc	Rib~Acc	Rib~Acc	DP~Acc	DP~Acc	DP~Acc	DP~Acc	DP~Acc	-
6	Rib~Acc	Rib~Acc	Rib~Acc	Rib~Acc	DP~Acc	DP~Acc	DP~Acc	DP~Acc	DP~Cau	-
7	Rib~Acc	Rib~Cau	Rib~Acc	Rib~Acc	DP~Cau	DP~Acc	DP~Cau	DP~Acc	DP~Cau	-
8	Rib~Cau	Rib~Acc	Rib~Acc	Rib~Acc	DP~Acc	DP~Acc	DP~Acc	DP~Acc	DP~Acc	-
9	Rib~Acc	Rib~Acc	Rib~Acc	Rib~Acc	DP~Acc	DP~Acc	DP~Acc	DP~Acc	DP~Acc	-
10	Rib~Acc	Rib~Acc	Rib~Acc	Rib~Acc	DP~Acc	DP~Acc	DP~Cau	DP~Acc	DP~Cau	-

IVDs: intervertebral discs, Cau: caudal articular process, Acc: accessary process, and DP: dorsal plane.

**Table 3 vetsci-11-00325-t003:** Safe corridor angles from the T10–11 to T13–L1 intervertebral discs.

No.	T10–11 IVDs		T11–12 IVDs		T12–13 IVDs		T13–L1 IVDs	
	S ^†^ (°)	F ^b^ (°)	S (°)	F (°)	S (°)	F (°)	S (°)	F (°)
1	8.35	37.43	7.45	34.47	9.36	33.25	8.46	39.42
2	7.83	34.52	9.72	35.12	7.48	33.55	9.62	41.53
3	9.23	36.68	10.53	39.78	8.34	33.53	*	37.35
4	5.71	37.74	14.34	41.39	14.45	32.48	10.63	36.53
5	5.41	33.87	5.35	40.27	15.58	38.76	11.47	41.34
6	7.85	30.58	10.15	40.36	8.51	38.86	4.48	40.00
7	5.95	37.58	8.19	35.62	8.75	40.47	3.19	38.56
8	6.96	32.75	8.41	34.23	8.43	43.63	9.89	39.53
9	8.79	30.35	10.43	36.54	10.43	33.89	10.35	35.61
10	4.79	32.84	12.63	34.73	14.62	42.95	*	41.35
Mean	7.09	34.43	9.72	37.25	10.60	37.14	8.51	39.12
SD	1.55	2.83	2.57	2.85	3.07	4.29	3.03	2.09

IVDs: intervertebral discs. S ^†^: the starting angle (the minimum value of the safe corridor angle). F ^b^: the finishing angle (the maximum value of the safe corridor angle). *: the starting safe corridor angle from the dorsal plane.

**Table 5 vetsci-11-00325-t005:** The safe corridor angle measurement ranges of the T10–11 to L6–7 intervertebral discs.

	20 J	40 J	60 J
Pre weight	3470.40 ± 228.57	4025.73 ± 181.09	3950.75 ± 212.18
Post weight	3452.15 ± 229.09	3998.97 ± 180.90	3912.25 ± 211.51
Difference weight	18.25 ± 2.15	26.77 ± 2.65	38.50 ± 2.49

**Table 4 vetsci-11-00325-t004:** Safe corridor angle from the L1–2 to L6–7 intervertebral discs.

No.	L1–2		L2–3		L3–4		L4–5		L5–6		L6–7	
	S ^a^ (°)	F ^b^ (°)	S (°)	F (°)	S (°)	F (°)	S (°)	F (°)	S (°)	F (°)	S (°)	F (°)
1	*	41.45	*	43.23	*	44.53	*	42.52	*	44.23	*	*
2	*	40.36	*	41.53	*	42.72	*	42.73	*	41.32	*	*
3	*	42.64	*	46.12	*	42.63	*	40.85	*	43.26	*	*
4	*	41.84	*	42.62	*	43.63	*	41.75	*	43.62	*	*
5	*	43.25	*	40.32	*	41.53	*	45.23	*	42.34	*	*
6	*	40.62	*	42.32	*	40.23	*	41.54	*	43.53	*	*
7	*	45.12	*	43.62	*	42.21	*	43.18	*	44.23	*	*
8	*	41.63	*	45.23	*	43.21	*	41.34	*	43.62	*	*
9	*	42.21	*	39.87	*	42.74	*	43.23	*	41.23	*	*
10	*	40.00	*	39.24	*	41.64	*	40.03	*	42.61	*	*
Mean		41.91		42.41		42.51		42.24		43.00		
SD		1.52		2.25		1.20		1.47		1.09		

S ^a^: the starting angle (the minimum value of the safe corridor angle). F ^b^: the finishing angle (the maximum value of the safe corridor angle). *: the starting safe corridor angle from the dorsal plane.

## Data Availability

The raw data supporting the conclusions of this article will be made available by the authors on request.

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
