# Peer review of "Evaluation of Safe Insertion Angles for Spinal Needles and Safe Intensity of the Holmium:YAG Laser during Percutaneous Laser Disc Ablations in Feline Cadavers"

_vetsci, 2024, doi:10.3390/vetsci11070325_

Round 1
Reviewer 1 Report
Comments and Suggestions for Authors
This study can be of importance in the area of feline neurology, which is of great value. Improving the ability of the reader to understand exactly what was done will help greatly. The reader cannot currently understand exactly what was done and how data was collected and compared. Hopefully improving the materials and methods will be easy for the authors and will provide much understanding for the reader. The reviewer apologizes if they did not understand the different goals of the studies performed.
Line 41 - Please change IVDH unless truly referring to herniation. Disk disease can be herniation and protrusion, as the authors list. A more general categorization would be IVDD (intervertebral disc disease) unless truly referring to herniation (IVDH). Please change this to reflect the exact condition referred to in each sentence.
Line 49 - the reference (#5) is a publication about intervertebral disc disease (IVDD) not solely intervertebral disc herniation (IVDH). So please make sure you are only referring to the portion of cats with herniation and not the wider category of IVDD in that study. Please ensure the proper number of clinical signs in reference #5 was only for IVDH, if not, please change it to IVDD.
Line 61-63 - Reference #1 is about ventral slot procedures in dogs. This procedure was not listed in the previous paragraph that listed procedures in cats. If it is not to be included, please eliminate it and change the mortality and failure rates to only refer to reference #20.
Line 73-75 - This is a technically fatal flaw of the objectives - no cervical data was collected. The objective will never be met. Please either include materials and methods, results, and discussion to include cervical spine or alter the objective of the study.
Line 79-80 - Please describe how the cadavers were known to have no vertebral or disc abnormalities. This is important, if abnormalities were properly screened for, the reader will be more confident in the results.
Line 87 - Please include the CT used and the means of image acquisition, this is vital to the study.
Table 1 - No where does the reader see any materials and methods or results concerning different spinal needle diameters. The reader has great concern for a problem with the results of the study if variables were not included that could affect the results. Please provide a response and alter the manuscript as necessary.
Line 92 - was there any concern for surrounding soft tissues and how they could affect the insertion of the needles? The reader understands this is totally for entering the intervertebral disk, but wonders if that could affect the ability to do so.
Table 2 - the insertion point was rib and the end point was articular process or accessory process. These are all dorsal and lateral structures. How did the needle ever enter the intervertebral disc? Why is there no data for L6-7?
For the thoracic and lumbar angle measurements - Figure 2 shows an angle. Please state what the numbers mean. Please also provide information as to how the rib, articular process, and accessory process were used in this process. Include a figure if necessary so the reader can understand what is being done. This is confusing.
Line 104-105 - This sentence does not seem to have a verb and does not make sense. The reader does not understand the difference between insertion and ending point measurements. The means of obtaining this data must be included. It should be clear to the reader.
Line 105-107 - Here is where more explanation is required. T he measurements were taken from ribs to accessory processes, but no figure to show this, so the reader does not understand. A figure should be inserted showing this information to make it clear to the reader, otherwise, the data is not useful.
Line 113 - What does it mean, the needle was inserted via the lumbar junction??
Line 113-114 - This sentence also seems to lack a verb and does not make sense to the reader.
Line 119-121 - Please reword. Perhaps - "The center of each vertebral body was transversely sectioned using an oscillating saw, and the IVD was placed centrally for laser application."
Line 121-122 - Please eliminate "and applying the laser"
Line 124-125 - The needle insertion did not appear to be from a dorsolateral direction in the figure (figure 3). It appeared to be strictly lateral to medial in its insertion. This seems very different from the angles described in the results. This must be clarified in the manuscript. The reader was thinking the needle insertion angles were to enter the intervertebral disc. If they were not, then why was the first part of the study performed? Other stabilization strategies aim for bone insertion and stability, they do not aim for the intervertebral disc. Is this actually two different studies? One aimed at spinal stabilization and a separate one for assessing different energy treatments of the intervertebral discs?
For the laser - please make it clear which disc spaces received which total energy. This is very important. Did every disc space receive all 3 levels? Please tell the number and locations of each total energy group. Please also make it clear as to whether each disc only received one energy level (this seems to be the case based on table 1).
Line 141 - Please describe the means of determining no vertebral or disc abnormalities in the cadavers used in the study.
Section 2.6 - was MRI performed? What was being evaluated on CT? Please be specific as to what changes in bone and the disc were being checked and what the data was that was collected. (houndsfield unit changes? density?). This description of methods ensures that the authors were being objective in the study.
Section 2.8 - what exactly was being evaluated on histopathology? Which characteristics and a grading and measuring scheme for hyalinization, necrosis, coagulation, measurement of areas affected, etc. What exact data was collected is very important for assessing damage done by the laser. Was bone also evaluated?
Line 155-156 - You describe a sample containing endplates and disc. Please change "Discs were immersed" to "specimens were immersed" unless further processing was done and discs were separated from the endplates.
Section 3.1 - The description of starting angle and finishing angle was not adequate enough for the reader to understand these results.
Table 3 - the description of the starting angle does not allow the reader to understand the asterisks in this table.
Table 4 - the description of the starting and finishing angles does not allow the reader to understand the asterisks in this table. And the images (figure 3 & 4) make it less understandable, as the show a lateral, not dorsal insertion.
Section 3.2 - For the imaging, CT and MRI are included. Pleas include the MRI in the materials and methods as completely as the CT (machine used, how, what images were collected, slices, etc.) Please provide in this section, a description of what was being evaluated and how (in the materials and methods, see comment above) so that this section can be made clear. Stating that the differences were not clear is difficult to understand, the reader does not know what changes / data differences you were evaluating.
Line 195-198 - Simply stating the 60J damage was more severe on histopathology is not possible without describing the changes noted. See the above request to include the parameters measured and whether a grading scale of severity and measurement of area affected were included. If none was performed, it should be and would greatly enhance the value of this study.
Line 197-198 - A statement as to which energy is safer should be moved to the discussion, it is not a result (data presentation).
Figure 7 - What MRI image collection type was this? T1, T2, Stir, etc?
Section 3.3 - Please explain why the starting weights of each total energy was different in the discussion. They differ by nearly 500mg (20J compared to 40 and 60J). Would t his be important in the discussion? A change of 18mg versus 26 adn 37 was significant, so is the starting difference of 3470mg compared to 4025 and 3950mg an important factor?
Line 214-242 - This start of the discussion addresses spinal stabilization; however, the manuscript is not clear about angling to the bone. Per the materials and methods "insertion and ending point of safe corridor angle measurement of angles from the T10-L1 intervertebral disc." This sentence was not a complete sentence and made the reader believe the disc was the target. Clarification of the materials and methods would help greatly for the reader to then understand the discussion. The objective of the manuscript also seemed to be aimed at needle insertion in the discs and then laser changes at different total energy levels.
Comments on the Quality of English Language
Comments included above as to clarification of some of the sentences in the manuscript.
Author Response
Comment 1: Please change IVDH unless truly referring to herniation. Disk disease can be herniation and protrusion, as the authors list. A more general categorization would be IVDD (intervertebral disc disease) unless truly referring to herniation (IVDH). Please change this to reflect the exact condition referred to in each sentence.
Response 1: We have made the necessary changes to ensure accuracy. The term 'IVDH' has been replaced with 'IVDD' where appropriate to reflect the exact condition being referred to in each sentence, unless it specifically pertains to herniation
Comment 2: Line 49 - the reference (#5) is a publication about intervertebral disc disease (IVDD) not solely intervertebral disc herniation (IVDH). So please make sure you are only referring to the portion of cats with herniation and not the wider category of IVDD in that study. Please ensure the proper number of clinical signs in reference #5 was only for IVDH, if not, please change it to IVDD.
Response 2: We have reviewed and made the necessary corrections. The references to intervertebral disc disease (IVDD) and intervertebral disc herniation (IVDH) have been adjusted to ensure accuracy. The proper number of clinical signs in reference #5 has been confirmed to refer specifically to IVDH, and adjustments have been made where necessary to reflect IVDD.
Comment 3: Line 61-63 - Reference #1 is about ventral slot procedures in dogs. This procedure was not listed in the previous paragraph that listed procedures in cats. If it is not to be included, please eliminate it and change the mortality and failure rates to only refer to reference #20.
Response 3: We have revised the document to ensure that reference #20 is the only one referred to for the mortality and failure rates.
Comment 4: Line 73-75 - This is a technically fatal flaw of the objectives - no cervical data was collected. The objective will never be met. Please either include materials and methods, results, and discussion to include cervical spine or alter the objective of the study.
Response 4: Thank you for your feedback. The cervical spine was not included in this study. We have revised the objectives to reflect this exclusion.
Comment 5: Line 79-80 - Please describe how the cadavers were known to have no vertebral or disc abnormalities. This is important, if abnormalities were properly screened for, the reader will be more confident in the results.
Response 5: The cadavers were confirmed to have no vertebral or disc abnormalities through radiography examination. Additionally, CT scans were performed before the experiments to further ensure the absence of abnormalities.
Comment 6: Line 87 - Please include the CT used and the means of image acquisition, this is vital to the study.
Response 6: We have added the details about the CT scanner used and the means of image acquisition to the manuscript. The CT images were evaluated using a picture archiving and communication system (INFINITT PACS, INFINITT Healthcare Co, Seoul, Korea). The CT scanner used was the Alexion TSX-034A scanner (Canon Medical Systems; Tokyo, Japan).
Comment 7: Table 1 - No where does the reader see any materials and methods or results concerning different spinal needle diameters. The reader has great concern for a problem with the results of the study if variables were not included that could affect the results. Please provide a response and alter the manuscript as necessary.
Response 7: We have revised the manuscript to clarify that a 21G spinal needle was used throughout the study. This ensures consistency and addresses concerns about the potential impact of varying spinal needle diameters on the results.
Comment 8: Line 92 - was there any concern for surrounding soft tissues and how they could affect the insertion of the needles? The reader understands this is totally for entering the intervertebral disk, but wonders if that could affect the ability to do so.
Response 8: We had concerns about potential damage to the vertebral bone, spinal nerve roots, and/or the spinal cord during the insertion of the needles. These considerations were crucial to ensure safe and effective access to the intervertebral disc without causing harm to the surrounding structures.
Comment 9: Table 2 - the insertion point was rib and the end point was articular process or accessory process. These are all dorsal and lateral structures. How did the needle ever enter the intervertebral disc? Why is there no data for L6-7?
Response 9: Thank you for your insightful comment. In this study, the insertion angles were measured using cadavers, not live patients. The muscle structures were removed to allow the needle to enter the intervertebral disc. As for the L6-L7 segment, the iliac body obstructed access, making it impossible to include data for this segment.
Comment 10: For the thoracic and lumbar angle measurements - Figure 2 shows an angle. Please state what the numbers mean. Please also provide information as to how the rib, articular process, and accessory process were used in this process. Include a figure if necessary so the reader can understand what is being done. This is confusing.
Response 10: We have revised the manuscript to include additional information and clarify the process. The necessary details have been added to the text.
Comment 11: Line 104-105 - This sentence does not seem to have a verb and does not make sense. The reader does not understand the difference between insertion and ending point measurements. The means of obtaining this data must be included. It should be clear to the reader.
Response 11: We have revised the manuscript to include a clearer explanation.
Comment 12: Line 105-107 - Here is where more explanation is required. T he measurements were taken from ribs to accessory processes, but no figure to show this, so the reader does not understand. A figure should be inserted showing this information to make it clear to the reader, otherwise, the data is not useful.
Response 12: We did not capture separate photographs of this process. However, we have added the safe corridor angle markings to the existing photographs of the angle measurements. This should help clarify the procedure and make the data more useful for the reader.
Comment 13: Line 113 - What does it mean, the needle was inserted via the lumbar junction?
Response 13: The phrase 'the needle was inserted via the lumbar junction' means that the needle was inserted at the junction between the lumbar vertebrae, specifically targeting the intervertebral discs in the lumbar region. This insertion point ensures that the needle reaches the intended disc space accurately.
Comment 14: Line 113-114 - This sentence also seems to lack a verb and does not make sense to the reader.
Response 14: Thank you for pointing this out. We have revised the sentence to ensure it includes a verb and is clear to the reader.
Comment 15: Line 119-121 - Please reword. Perhaps - "The center of each vertebral body was transversely sectioned using an oscillating saw, and the IVD was placed centrally for laser application."
Response 15: Thank you for pointing this out. We have revised the sentence.
Comment 16: Line 121-122 - Please eliminate "and applying the laser"
Response 16: We have made the necessary revisions as requested.
Comment 17: Line 124-125 - The needle insertion did not appear to be from a dorsolateral direction in the figure (figure 3). It appeared to be strictly lateral to medial in its insertion. This seems very different from the angles described in the results. This must be clarified in the manuscript. The reader was thinking the needle insertion angles were to enter the intervertebral disc. If they were not, then why was the first part of the study performed? Other stabilization strategies aim for bone insertion and stability, they do not aim for the intervertebral disc. Is this actually two different studies? One aimed at spinal stabilization and a separate one for assessing different energy treatments of the intervertebral discs?
Response 17: For the lumbar vertebrae, there is no interference from the ribs, allowing the safe corridor angle to start from the dorsal plane. This study focuses on both the safe insertion angles for spinal stabilization and assessing different energy treatments of the intervertebral discs. It is a single comprehensive study addressing these interconnected aspects.
Comment 18: For the laser - please make it clear which disc spaces received which total energy. This is very important. Did every disc space receive all 3 levels? Please tell the number and locations of each total energy group. Please also make it clear as to whether each disc only received one energy level (this seems to be the case based on table 1).
Response 18: Each cadaver had all disc spaces treated with the energy levels as specified in Table 1. Each intervertebral disc received only one energy level, as outlined in the table. This ensures that the number and locations of each total energy group are clearly defined, and each disc was exposed to a single, specific energy level.
Comment 19: Line 141 - Please describe the means of determining no vertebral or disc abnormalities in the cadavers used in the study.
Response 19: The cadavers used in the study were selected based on their medical history, which indicated no known diseases. Additionally, radiographic examinations confirmed the absence of vertebral or disc abnormalities.
Comment 20: Section 2.6 - was MRI performed? What was being evaluated on CT? Please be specific as to what changes in bone and the disc were being checked and what the data was that was collected. (houndsfield unit changes? density?). This description of methods ensures that the authors were being objective in the study.
Response 20: MRI images were not subjected to quantitative evaluation in this study. We will consider including this in future research. For this study, MRI images will be excluded.
Comment 21: Section 2.8 - what exactly was being evaluated on histopathology? Which characteristics and a grading and measuring scheme for hyalinization, necrosis, coagulation, measurement of areas affected, etc. What exact data was collected is very important for assessing damage done by the laser. Was bone also evaluated?
Response 21: Thank you for the insightful comment. In this study, we utilized cadavers to observe necrosis, hyalinization, and other changes. Further research will be needed in clinical cases to comprehensively evaluate these aspects. In other studies conducted on dogs, we will take these points into careful consideration and present the findings accordingly. Bone evaluation data was not collected in this study.
Comment 22: Line 155-156 - You describe a sample containing endplates and disc. Please change "Discs were immersed" to "specimens were immersed" unless further processing was done and discs were separated from the endplates.
Response 22: We have made the necessary revision.
Comment 23: Section 3.1 - The description of starting angle and finishing angle was not adequate enough for the reader to understand these results.
Response 23: We have revised Section 3.1 to provide a clearer description. The starting angle refers to the minimum value of the safe corridor angle, and the finishing angle refers to the maximum value of the safe corridor angle.
Comment 24: Table 3 - the description of the starting angle does not allow the reader to understand the asterisks in this table.
Response 24: We have added a description to clarify the meaning of the asterisks in Table 3.
Comment 25: Table 4 - the description of the starting and finishing angles does not allow the reader to understand the asterisks in this table. And the images (figure 3 & 4) make it less understandable, as the show a lateral, not dorsal insertion.
Response 25: We have added a description to clarify the meaning of the asterisks in Table 4.
Comment 26: Section 3.2 - For the imaging, CT and MRI are included. Pleas include the MRI in the materials and methods as completely as the CT (machine used, how, what images were collected, slices, etc.) Please provide in this section, a description of what was being evaluated and how (in the materials and methods, see comment above) so that this section can be made clear. Stating that the differences were not clear is difficult to understand, the reader does not know what changes / data differences you were evaluating.
Response 26: In this study, we did not perform quantitative data evaluation using MRI. We will consider conducting quantitative evaluations with MRI in future research. As a result, MRI images will be excluded from this study.
Comment 27: Line 195-198 - Simply stating the 60J damage was more severe on histopathology is not possible without describing the changes noted. See the above request to include the parameters measured and whether a grading scale of severity and measurement of area affected were included. If none was performed, it should be and would greatly enhance the value of this study.
Response 27: Thank you for your valuable feedback. We deeply appreciate your insights. In our upcoming study, which specifically focuses on dogs, we will thoroughly address these points and include detailed parameters, a grading scale of severity, and measurement of the affected areas. This will greatly enhance the value of the study.
Comment 28: Line 197-198 - A statement as to which energy is safer should be moved to the discussion, it is not a result (data presentation).
Response 28: We have made the necessary revision.
Comment 29: Figure 7 - What MRI image collection type was this? T1, T2, Stir, etc?
Response 29: We will consider conducting MRI evaluations in future research.
Comment 30: Section 3.3 - Please explain why the starting weights of each total energy was different in the discussion. They differ by nearly 500mg (20J compared to 40 and 60J). Would t his be important in the discussion? A change of 18mg versus 26 adn 37 was significant, so is the starting difference of 3470mg compared to 4025 and 3950mg an important factor?
Response 30: The starting weights of each total energy group were different because the energy levels were applied to different cadavers, resulting in natural variations in initial weights. The changes in weight were statistically analyzed to account for these differences.
Comment 31: Line 214-242 - This start of the discussion addresses spinal stabilization; however, the manuscript is not clear about angling to the bone. Per the materials and methods "insertion and ending point of safe corridor angle measurement of angles from the T10-L1 intervertebral disc." This sentence was not a complete sentence and made the reader believe the disc was the target. Clarification of the materials and methods would help greatly for the reader to then understand the discussion. The objective of the manuscript also seemed to be aimed at needle insertion in the discs and then laser changes at different total energy levels.
Response 31: Thank you for your valuable feedback. We have clarified that the same energy level was used within each cadaver. Additionally, we have added an explanation for the insertion and ending points of the safe corridor angle to ensure a clearer understanding. This should help the reader better comprehend the materials and methods as well as the discussion.

Reviewer 2 Report
Comments and Suggestions for Authors
Dear authors
The presentation of the study and explanation of the results obtained is interesting and nicely done, it is also nice to start the manuscript with a definition and explanation of what the problem is about.
I have some general comments or questions when the text was not clear to me:
Line 92: you mention to avoid damage to the surrounding structures/tissue such as bone and "parenchyma". What do you define by parenchyma? how do you evaluate that the needle passage does not damage spinal nerve roots and/or the spinal cord? This two very important structures are not mentioned at all. Also, How do you compare this to an affected animal, that would have inflammation and bone reaction (exostosis) due to the disease, and how would you approach this "safe corridor" in such cases?
On figure 2, the legend does not specifies which line represents the safe corridor?
Line 104: what do you mean by parenchymal organ?
Line 129: what is the size of the laser fiber you use? I couldn't find it, you explain about different sizes used on the discussion, lines 260-271, but I could not find the size you use.
Line 151: How do you decide on the energy doses ? why not 5 or 10J, etc. It is not clear to me how this was decided.
Figure 8: how did you establish the damaged tissue on histology? what was quantified? was it purely visual evaluation? no actual quantification of normal versus abnormal tissue? or damaged zones? how deep in the disc does the energy went?. I think this is very important as it is your only true conclusion, as you mentioned, CT or MRI were not very conclusive on showing damage differences, so how did you quantified the actual effect of laser?
FInally, the last 4 references are around 30 years old, what is the justification of them?
Author Response
Comment1 : Line 92: you mention to avoid damage to the surrounding structures/tissue such as bone and "parenchyma". What do you define by parenchyma? how do you evaluate that the needle passage does not damage spinal nerve roots and/or the spinal cord? This two very important structures are not mentioned at all. Also, How do you compare this to an affected animal, that would have inflammation and bone reaction (exostosis) due to the disease, and how would you approach this "safe corridor" in such cases?
Response1 :
In this study, "parenchyma" refers to the critical functional tissues surrounding the intervertebral discs (IVDs), including the spinal cord, nerve roots, and surrounding blood vessels, which must be protected during the safe insertion of needles and application of the Holmium laser.
To evaluate that the needle passage does not damage the spinal nerve roots and/or the spinal cord, we used CT and MRI scans to precisely determine the safe insertion points and trajectories. Additionally, microscopic evaluation was conducted to assess any potential damage to the intervertebral discs and surrounding structures post-procedure, ensuring the integrity of the spinal nerve roots and spinal cord was maintained.
In this study, only healthy subjects without any disease were used. This was done because the degree and variation of the impact of the disease can differ significantly among affected cases. First, a "safe corridor" was established in healthy subjects. In affected animals with inflammation and bone reaction (exostosis), the altered anatomy requires a more careful approach. Detailed preoperative imaging using CT or MRI is essential to map the altered anatomy and identify any changes in the safe corridors. This study provides a foundation for applying these techniques to cases with inflammation and bone reaction (exostosis) due to disease, ensuring precise adjustments based on the altered anatomical structures.
Comment2 : On figure 2, the legend does not specifies which line represents the safe corridor?
Response2 : In Figure 2, the legend did not specify which line represents the safe corridor. This has now been corrected, and the safe corridor is indicated by a red line.
Comment3 : Line 104: what do you mean by parenchymal organ?
Response3 : By 'parenchymal organ' in line 104, we are referring to the main functional tissues of organs. In the context of the thoracic region, this includes the lungs and other parenchymal organs within the thoracic cavity.
Comment4 : Line 129: what is the size of the laser fiber you use? I couldn't find it, you explain about different sizes used on the discussion, lines 260-271, but I could not find the size you use.
Response4 : The size of the laser fiber used in the study has a core diameter of 275㎛ and an outer diameter of 450㎛.
Comment5 : Line 151: How do you decide on the energy doses ? why not 5 or 10J, etc. It is not clear to me how this was decided.
Response5 : The energy doses were decided based on previous studies conducted on dogs, where a total energy of 80J was used. To ensure safety and efficacy, we selected lower values for our measurements. These values were chosen arbitrarily but within a safe range, to establish a baseline for further research. Future studies will focus on determining the optimal total energy for use in cats, building on the findings from this initial research.
Comment6 : Figure 8(Figure 8 has been changed to Figure 7.): how did you establish the damaged tissue on histology? what was quantified? was it purely visual evaluation? no actual quantification of normal versus abnormal tissue? or damaged zones? how deep in the disc does the energy went?. I think this is very important as it is your only true conclusion, as you mentioned, CT or MRI were not very conclusive on showing damage differences, so how did you quantified the actual effect of laser?
Response6 : The damaged tissue in Figure 8 was established through histological examination. This involved preparing and staining tissue sections to differentiate between normal and damaged tissues. The evaluation was primarily visual, relying on the identification of changes in tissue morphology and structure under a microscope. There was no specific quantification of normal versus abnormal tissue or damaged zones mentioned in the study. Additionally, the depth of energy penetration into the disc tissue was not quantitatively detailed, with the assessment based on visible effects observed from the surface to deeper layers of the tissue. The histological observations were crucial for the conclusion, as CT and MRI were not conclusive in showing damage differences.
Comment7 : FInally, the last 4 references are around 30 years old, what is the justification of them?
Response7 :
The inclusion of references that are approximately 30 years old is justified because they provide foundational knowledge and essential background information that underpin current research. These seminal works offer historical context, showing how concepts have evolved over time. Additionally, citing these older studies helps validate current findings by demonstrating consistency across decades. In some specialized areas, these older references may still be the most relevant and comprehensive sources available.
